# DDX5 Functions as a Tumor Suppressor in Tongue Cancer

**DOI:** 10.3390/cancers15245882

**Published:** 2023-12-18

**Authors:** Qingqing Liu, Yangqing Sun, Min Long, Xueyan Chen, Shangwei Zhong, Changhao Huang, Rui Wei, Jun-Li Luo

**Affiliations:** 1Department of Oncology, Xiangya Hospital, Central South University, Changsha 410008, China; liuqq_2018@csu.edu.cn (Q.L.); 198101050@csu.edu.cn (Y.S.); chenxueyan725@gmail.com (X.C.); 2The Cancer Research Institute and the Second Affiliated Hospital, Hengyang Medical School, University of South China, Hengyang 421001, China; 20221013110075@stu.usc.edu.cn (M.L.); swzhong@usc.edu.cn (S.Z.); 3The Organ Transplant Center, Xiangya Hospital Central South University, Changsha 410000, China; drhuangchanghao@hotmail.com; 4National Health Commission Key Laboratory of Birth Defect Research and Prevention, Hunan Provincial Maternal and Child Health Care Hospital, Changsha 410008, China

**Keywords:** DDX5, tumor suppressor, tongue cancer, CD8+ T cell, macrophage, immune cell infiltration

## Abstract

**Simple Summary:**

In this study, we demonstrate that DDX5 serves as a tumor suppressor in the specific context of tongue cancer, which is contrary to its documented oncogenic role in a vast array of cancers. The high expression of DDX5 in tongue cancer is correlated with a better prognosis in clinical patients. The knockdown of DDX5 promotes, while the overexpression of DDX5 inhibits, tongue cancer proliferation, development, and cisplatin resistance. Moreover, we found that the expression of DDX5 in tongue cancer is associated with immune cell infiltration in the tumor microenvironment. Specifically, the expression of DDX5 is associated with a reduced infiltration of M2 macrophages and an increased infiltration of T cell clusters, which may contribute to anticancer effects in the tumor microenvironment. Our study establishes DDX5 as a valuable prognostic biomarker and an important tumor suppressor in tongue cancer.

**Abstract:**

DEAD-box polypeptide 5 (DDX5), a DEAD-box RNA helicase, is a multifunctional protein that plays important roles in many physiological and pathological processes. Contrary to its documented oncogenic role in a wide array of cancers, we herein demonstrate that DDX5 serves as a tumor suppressor in tongue cancer. The high expression of DDX5 is correlated with better prognosis for clinical tongue cancer patients. DDX5 downregulates the genes associated with tongue cancer progression. The knockdown of DDX5 promotes, while the overexpression of DDX5 inhibits, tongue cancer proliferation, development, and cisplatin resistance. Furthermore, the expression of DDX5 in tongue cancer is associated with immune cell infiltration in the tumor microenvironment. Specifically, the expression of DDX5 is associated with the reduced infiltration of M2 macrophages and increased infiltration of T cell clusters, which may contribute to anticancer effects in the tumor microenvironment. In this study, we establish DDX5 as a valuable prognostic biomarker and an important tumor suppressor in tongue cancer.

## 1. Introduction

Head and neck squamous cell carcinomas (HNSCCs) encompass carcinomas of the hypopharynx, larynx, oropharynx, nasopharynx, and oral cavity. There are various types of oral carcinomas, including those of the tongue, the hard palate, the lower and upper alveolar ridge, the floor of the mouth, and the retromolar trigone and the mucosa of the buccal cavity [1]. Among these, tongue cancer stands out as one of the most prevalent. A significant number of tongue cancer cases are identified at advanced stages, resulting in a severely reduced 5-year survival rate for affected patients [2]. The poor prognosis of tongue cancers is primarily due to tumor aggressiveness, early metastasis, and therapy resistance [3]. Therefore, identifying the molecular mechanisms underlying these tumor characteristics improves the understanding of tongue cancer progression and facilitates the development of novel therapeutic strategies.

DEAD-box polypeptide 5 (DDX5) belongs to the DEAD-box RNA helicase family, the members of which utilize the energy derived from ATP hydrolysis to unwind double-stranded RNA (dsRNA) molecules [4]. DDX5 is a multifunctional protein and participates in numerous cellular processes, including mRNA splicing, mRNA export and translation, and ribosome and miRNA bioDDX5sis [5]. The current literature underscores the oncogenic role DDX5 plays in almost all types of cancers, including lung, colon, and breast cancers [6,7,8]. DDX5 plays important roles in maintaining genome stability [9], especially in the context of DNA damage response. The phosphorylation of DDX5 at residues Y593 and Y595 inhibits TRAIL-induced apoptosis and promotes tumor cell proliferation, metastases, and epithelial–mesenchymal transition (EMT) [6]. Essentially, the dysregulation of DDX5 in various cancer types substantiates its role in facilitating cancer cell proliferation, even though it might assume tumor suppressor functions in specific contexts [6].

In this study, we show that DDX5 functions as a tumor suppressor and may serve as a valuable prognostic biomarker in tongue cancer. Elevated DDX5 expression in tongue cancer is associated with a more favorable prognosis for clinical patients. The knockdown of DDX5 accelerates, while its overexpression inhibits, tongue cancer proliferation, development, and resistance to cisplatin. Moreover, DDX5 expression in tongue cancer is associated with immune cell infiltration in the tumor microenvironment.

## 2. Materials and Methods

### 2.1. Cell Culture

SSC-9 and Cal27 cells, obtained from Procell (Wuhan, China), were cultured in DMEM/F-12 (Hyclone, Logan, UT, USA) plus 1% penicillin–streptomycin (Biosharp, Shanghai, China), 0.4% hydrocortisone, and 10% FBS, or in DMEM (Hyclone) plus 1% penicillin–streptomycin and 10% FBS.

### 2.2. Plasmid and Lentivirus Preparation

For the construction of the DDX5 overexpression plasmid, LVCV-19 vector, the DDX5-CDS was amplified from DDX5 cDNA plasmid (Sino Biological, Inc., Beijing, China) and the sequences of primers used were as follows: forward primer: ATGTCGGGTTATTCGAGTGAC; reverse primer: TTAGGCGTAGTCAGGCACGTC (with HA tag). For the construction of DDX5 knockdown plasmid, pLKO.1 vector, the forward primer was 5′-CCGGGCTCCTATTCTGATTGCTACACTCGAGTGTAGCAATCAGA ATAGGAGCTTTTT-3′ and the reverse primer was 5′-AATTAAAAAGCTCCTATTCTGATTGCTACACTCGAGTGTAGCAATCAGAATAGGAGC-3′. The DDX5 overexpression (LVCV-19-DDX5) or DDX5 knockdown plasmid (pLKO.1-DDX5) plus packaging plasmids (pMD2G and psPAX2) were transfected into 293T cells. Then, 48 h later, the supernatant was collected to harvest the lentiviral particles. The primers for the construction of the shRNA plasmid against MMP10 mRNA were as follows: forward primer: 5′-CCGGCCTGGGCTTTATGGAGATATTCTCGAGAATATCTCCATAAAGCCCAGGTTTTTG-3′; reverse primer: 5′-AATTCAAAAACCCTCACTCCTCT CCTAATTACTCGAGTAATTAGGAGAGGAGTGAGGG-3′.

### 2.3. Establishment of Stable Tongue Cancer Cell Lines

The tongue cancer cells were infected with DDX5 shRNA or overexpression lentivirus. Then, the cells were seeded in 10 cm cell culture dishes 48 h later, and cultured in complete medium with puromycin (Sangon Biotech, Shanghai, China). Single colonies were collected and cultured in 96-well plates. The efficiency of DDX5 in the overexpression and knockdown clones was analyzed by Western blot.

### 2.4. Cell Proliferation and Colony Formation Analysis

For the analysis of cell viability and proliferation rates, a Cell Counting Kit-8 (CCK-8) assay was applied. Briefly, 96-well plates were used to culture the cells under the indicated conditions. Then, CCK-8 solution (Beyotime, Shanghai, China) was added to the wells and incubated at 37 °C. The plates were measured 3 h later at 450 nm using a microplate reader (Molecular Devices, San Jose, CA, USA). To examine the effect of DDX5 on the cell colony formation, tongue cancer cells were seeded in 6-well plates and cultured under the indicated conditions. The cell colonies were washed with PBS and fixed with 80% methanol for 10 min after the clones had more than 50 cells. Then, 0.1% crystal violet (Sigma) was used to stain the cell colonies, and the number of colonies was counted.

### 2.5. Wound Healing Assay

When the cells seeded in 12-well plates reached 100% confluence, a 20 μL pipette tip was used to scratch the cell monolayers and then they were gently washed with PBS to remove the cellular debris. A phase-contrast microscope was applied to take the cell images at different time points. Cell migration rate (%) = (wound distance at 0 h–wound distance at indicated time point)/wound distance at 0 h × 100%, where 0 h means the time of cell scratch.

### 2.6. Cisplatin Treatment

Cisplatin (QILU Pharmaceutical) was used to treat cells at different concentrations (0, 2, 4, 8, and 16 μM) in 96-well plates or 6-well plates for 48 h. Cell viability was detected with a CCK-8 assay. Cell viability rate (%) = OD_450_ value (cells treated with indicated cisplatin concentration)/OD_450_ value (cells treated with vehicle) × 100%. A series of cisplatin concentrations were used to treat the cells seeded in 6-well plates until colonies formed. Then, 0.1% crystal violet (Sigma, St. Louis, MO, USA) was applied to stain the colonies, and the number of cell colonies formed in each group was counted. The control group comprised cells treated with the vehicle.

### 2.7. Real-Time PCR

For real-time PCR, firstly, a TAKARA reverse transcription kit (Takara, Beijing, China) was used to perform the reverse transcription according to the manufacturer’s instructions. Then, a PCR master mix (Sango Biotech, Shanghai, China) was used, and the detection was performed by using Bio-Rad CFX Maestro (Hercules, CA, USA). The sequences of primers used were as follows: MMP10-F: 5′-TCAGTCTCTCTACGGACCTCC-3′, MMP10-R: 5′-CAGTGGGATCTTCGCCAAAAATA-3′; DDX5-F: 5′-AGAGAGGCGATGGGC CTATTT-3′, DDX5-R: 5′-CTTCAAGCGACATGCTCTACAA-3′. The internal control was GAPDH, and its primers were GAPDH-F: 5′-CTGGGCTACACTGAGCACC-3′ and GAPDH-R: 5′-AAGTGGTCGTTGAG GGCAATG-3′.

### 2.8. Western Blot Analysis

The cell lysates were separated via SDS-PAGE gel and then transferred to a PVDF membrane (Millipore, Billerica, MA, USA). After being incubated with blocking solution containing 5% non-fat dry milk at room temperature for 2 h, the membrane was incubated with primary antibodies DDX5 (anti-DDX5 antibody, CST, 1:1000) and GAPDH (anti-GAPDH antibody, Proteintech Technology, 1:1000) overnight at 4 °C. On the next day, the corresponding HRP-conjugated secondary antibody (Abbkine, Wuhan, China) was used to incubate the membrane at room temperature for 1 h. An ECL kit (Millipore) was used to detect chemiluminescence.

### 2.9. Xenograft Tumor Mouse Models

For the mouse models, 6-week-old NOD-SCID mice (SJA Laboratory Animal Ltd., Changsha, China) were used. In total, 5 × 10^6^ Cal 27 cells were subcutaneously inoculated to generate the xenograft tumors in mice. Tumor development was monitored and measured every three days, and the formula V = ab^2^/2 was applied to calculate the tumor volume, where ‘a’ means the greatest diameter and ‘b’ means the perpendicular diameter. The animal experiments were approved by the Xiangya Medical School’s Institutional Animal Care and Use Committee.

### 2.10. Human Tongue Cancer Samples

In total, 169 formalin-fixed and paraffin-embedded tongue cancer tissue samples were included in our study, and encoded clinicopathological data and follow-up information about these samples were collected from tongue cancer patients treated at the Xiangya Hospital, Central South University, between June 2010 and December 2016. Our study was approved by the Research Ethics Committee of Xiangya Hospital. The clinicopathological parameters of patients with tongue cancer are provided in Table 1.

### 2.11. Immunohistochemical and Immunofluorescence Staining

Immunohistochemical staining was performed to detect DDX5 expression in tongue cancer samples. Briefly, the tongue cancer tissue sections (4 μm thick) were deparaffinized and rehydrated. Then, the Tris-EDTA (pH 9.0) buffer was used for antigen retrieval. After being blocked with 3% H_2_O_2_ solution for 10 min, the sections were blocked with 5% goat serum for 1 h, and the slides were incubated with DDX5 primary antibodies (Abcam, ab126730) overnight at 4 °C [10]. On the next day, the slides were treated with Two-Step IHC reagents and 3,3-diaminobenzidine (DAB) solution according to the manufacturer’s instructions. Harris’ modified hematoxylin was used in counterstaining. The staining intensity in samples was negative (0), weak (1), moderate (2), and strong (3), and the percentage of positive cells was defined as 0 (0–5%), 1 (5–25%), 2 (26–50%), 3 (51–75%), or 4 (>75%). The multiplication of the staining intensity and positive cell scores was applied to calculate the total score for each sample. The sample was defined as high DDX5 expression when the total score was ≥5, while it was classified as low DDX5 expression if the total score was <5. For the immunofluorescence (IF) assay, secondary antibodies (AF488, AF594, Thermo Fisher Scientific, Waltham, MA, USA) were used.

### 2.12. RNA Sequencing and Analysis

Total RNA was extracted from DDX5 knockdown Cal27 and control cells using an RNeasy Plus Mini Kit (QIAGEN) following the manufacturer’s protocol. Following quality control, high-quality total RNA samples with an Agilent Bioanalyzer RIN > 7.0 were employed in the construction of the RNA sequencing library. An RNA sequencing study was carried out on DDX5 knockdown Cal27 and control cells using a BGISEQ-500 (BGI Company, Shenzhen, China). SOAPnuke (v1.5.2) was used to filter the raw data [11], and the clean data were mapped onto the reference genome by HISAT (v2.1.0) [12]. The assembled unique DDX5 was mapped from the clean data by using Bowtie2 (v2.2.5) [13]. The RSEM (v1.2.8) was applied to calculate the expression level of genes [14]. The DESeq was used to perform the within-group differential DDX5 analysis [15] under the conditions of adjusted *p* value ≤ 0.001 and fold change ≥ 2. Poisson distribution was obtained by using between-group differential DDX5 analysis under the conditions of FDR ≤ 0.001 and fold change ≥ 2.

### 2.13. Data Collection and Procession

Single-cell RNA sequencing (scRNA-seq) data GSE172577 [16] were obtained from the GEO database and included scRNA-seq data (10X Genomics) of 6 cases of tongue cancer tissues. The clinical details and expression profiles in the GSE193445 [17], GSE164619 [18], and GSE30784 [19] datasets were derived from the GEO database (https://www.ncbi.nlm.nih.gov/geo/, accessed on 18 March 2022).

The processing of scRNA-seq data took place as follows: (1) The Seurat package was applied to preprocess the scRNA-seq data, the proportion of mitochondrial genes was determined by the PercentageFeatureSet function, and the relationship between sequencing depth and mitochondrial gene sequences and/or total intracellular sequences was investigated using correlation analysis. (2) Each gene to be expressed in ≥5 cells was set. (3) The expression of genes in each cell was >300 and <5000, the UMI of each cell was ≥1000, and the content of mitochondria was <10%. (4) To normalize the scRNA-seq data following data filtering, the LogNormalize method was applied.

The processing of microarray data of the GEO tongue cancer cohort was as follows: (1) The probe IDs were converted to gene symbol format. (2) For their correspondences to multiple genes, probes were removed. (3) The gene expression was regarded as the average value, and multiple probes corresponded to one gene.

### 2.14. Differential Expression Analysis, Gene Set Enrichment Analysis (GSEA), and Gene Set Variation Analysis (GSVA)

Based on DDX5 median expression (lower than 50% and higher than 50%), the samples of each cancer type derived from the TCGA database were classified into two groups. To evaluate the activities of these pathways, single-sample gene set enrichment analysis (GSEA) in R was applied. The “c2.cp.kegg.v7.4.symbols.gmt” file from the MSigDB was used to analyze the relationship between DDX5 and splicing pathways.

### 2.15. Identification of Malignant Cells via Copy Number Variation (CNV)

In order to identify malignant cells from epithelia, the CopyKAT algorithm (version 1.1.0) [20] was applied to estimate the genomic copy number profiles and identify aneuploid tumor cells. The sum of the calculated CNV for each gene per cell was the CNV score of the cell.

### 2.16. Monocle 2

Monocle 2 (version 2.14.0) was used to infer the cell lineage trajectory of T cells, DCs, neutrophils, and myeloid cells with the top 1000 signature genes with a *p* value < 0.001, calculated using the differential gene test function. After dimension reduction and cell ordering, the differentiation trajectory was inferred with the default parameters of Monocle 2.

### 2.17. Drug Sensitivity Analysis

The gene expression data (RPKM matrix) of the tongue cancer cell lines were derived from the CCLE database (https://portals.broadinstitute.org/ccle/, accessed on 6 June 2022), and the IC50 data of drugs were obtained from the Genomics of Drug Sensitivity in Cancer (GDSC) database (https://www.cancerrxgene.org, accessed on 6 June 2022). To predict the chemotherapeutic response of the DDX5-high and DDX5-low groups of the tongue cancer cells, the pRRophetic package was used. The IC50 of tongue cancer was estimated by using ridge regression analysis, and the prediction accuracy was evaluated using 10-fold cross-validation.

### 2.18. Analysis of the Link between DDX5 Expression in Tumor Cells and Infiltrated Immune Cells in the Tumor Microenvironment 

The link between DDX5 expression in cancer cells and the immune cell infiltration in the TME was systemically analyzed. The TCGA database was used to create precise immunological features of tumor-infiltrating cells in the indicated tumor types based on the CIBERSORT algorithm. The link between DDX5 expression and the quantity of infiltrated immune cells, including CD4+ T cells, CD8+ T cells, neutrophils, B cells, macrophages, and dendritic cells (DCS), was also examined.

### 2.19. Statistical Analysis

For the graphing and statistical analyses, Prism 9.5.0 (GraphPad Software, San Diego, CA, USA) was used. Either t-tests or analyses of variance (ANOVAs) were used to establish statistical significance, as appropriate. Data show the mean ± the standard error of the mean (SEM).

## 3. Results

### 3.1. High Expression of DDX5 in Tongue Cancer Is Associated with Better Clinical Prognosis

To investigate the association of DDX5 with human tongue cancer development, immunohistochemistry (IHC) was applied to analyze the DDX5 protein level in 169 cases of paraffin-embedded human primary tongue cancer tissues (Figure 1A). Our data indicated that high DDX5 expression in tumor tissue was associated with a lower rate of relapse within 3 years after surgery and chemo/radiotherapy, as well as a better overall survival (OS) when compared to those with low DDX5 expression (Figure 1B,C), suggesting that DDX5 may play an important role in the development of tongue cancer.

We further analyzed the expression of DDX5 in tongue cancer in three datasets (GSE193445, GSE164619, GSE30784). We found that, in the context of tongue cancer treatment, tumors with high DDX5 expression exhibited partial to complete remission. Conversely, a substantial downregulation of DDX5 was observed in cases of advanced tongue cancer (Appendix A). DDX5 expression increases as normal tongue tissue transitions into precancerous lesions, but it is markedly downregulated in the transformation into cancerous tissue (Appendix A–D). Furthermore, we conducted an in-depth analysis to delineate the correlation between DDX5 and signatures associated with cancer immunotherapy and cell cycles (Appendix A).

### 3.2. DDX5 Knockdown Promotes Tongue Cancer Cell Proliferation and Mobility

To delineate the role of DDX5 in tongue cancer, we established tongue cancer cell lines with a stable knockdown of DDX5 (DDX5-KD). This was accomplished by infecting SCC-9 and Cal 27 cells with lentiviruses designed to express DDX5 shRNA that targets the human DDX5 CDS, followed by selection with puromycin. The knockdown efficiency of DDX5 in these cells was validated through Western blot analysis (Figure 1D). Subsequently, we investigated the proliferation, colony formation, and migration capacities of these cells by utilizing a CCK-8 assay (Figure 1D), cell colony formation assay (Figure 1E), and scratch wound healing assay (Figure 1F), respectively. Our data revealed that DDX5 knockdown significantly augmented the proliferation (Figure 1D), colony formation (Figure 1E), and migration and mobility (Figure 1F) of SCC-9 and Cal 27 cells when compared with their respective control cells.

### 3.3. DDX5 Knockdown Promotes Tongue Cancer Xenograft Tumor Development

To examine whether the knockdown of DDX5 promotes tongue cancer development in mouse models, 5 × 10^6^ DDX5-KD or control Cal 27 cells were injected subcutaneously (s.c.) into the flanks of 6-week-old NOD-SCID mice, and tumor growth and development in these mice were monitored every three days. We found that the knockdown of DDX5 significantly promoted tongue cancer xenograft tumor development in mouse models (Figure 1G–I).

### 3.4. Overexpression of DDX5 Inhibits Tongue Cancer Cell Proliferation and Mobility

To further examine the effect of DDX5 expression on tongue cancer cells, the stable DDX5-overexpression (DDX5-OE) tongue cancer cell lines were established by infecting SSC-9 and Cal 27 cells with lentiviruses engineered to overexpress human DDX5, followed by puromycin selection. Western blot analysis was used to confirm the efficiency of DDX5 overexpression in these cells compared with the control cells (Figure 2A). The CCK-8, scratch wound healing assays, and cell colony formation were used to analyze the effect of DDX5 overexpression on the proliferation, colony formation, and migration of tongue cancer cells. We found that DDX5 overexpression significantly reduced cell proliferation (Figure 2A), colony formation (Figure 2B), and migration and mobility (Figure 2C) in both SCC-9 and Cal 27 cells.

### 3.5. DDX5 Overexpression Suppresses Tongue Cancer Xenograft Tumor Development

Mouse models were used to examine the effect of DDX5 overexpression on tongue cancer development. Briefly, 2 × 10^6^ control or DDX5-OE Cal 27 cells were injected s.c. into the flanks of 6-week-old NOD-SCID mice, and tumor growth and development in these mice were monitored. Consistently, the result of tongue cancer xenograft mouse models showed that the overexpression of DDX5 significantly suppressed tumor growth and development (Figure 2D–F).

### 3.6. Knockdown of DDX5 Upregulates the DDX5s Associated with Tongue Cancer Progression

To uncover the mechanisms through which DDX5 inhibits tongue cancer progression, we performed a comparative analysis of DDX5 expression profiles between control and DDX5-KD Cal 27 cells utilizing RNA sequencing analysis. The volcano plot and heatmap clearly delineated the differentially expressed DDX5s (DEGs) between control and DDX5-KD Cal 27 tongue cancer cells (Figure 3A,B). Subsequently, we employed gene ontology (GO) and gene set enrichment analysis (GSEA) to scrutinize the DEGs further. Intriguingly, we discovered that the genes upregulated in DDX5-KD tongue cancer cells are chiefly involved in the positive regulation of cytokine production, the regulation of viral processes, and the negative regulation of innate immune response. In contrast, the downregulated genes primarily associate with the regulation of the mitotic cell cycle, DNA replication, the P53 signaling pathway, and mitotic sister chromatid segregation (Figure 3C,D). This evidence starkly demonstrates that DDX5 knockdown escalates the expression of genes promoting cancer progression while suppressing those maintaining cell homeostasis and tumor suppression.

Furthermore, we conducted a detailed analysis of some of these differentially expressed genes (DEGs) in both DDX5-KD and DDX5-OE Cal 27 and SCC-9 cells. In particular, we noticed a marked upregulation of MMP10 expression in DDX5-KD Cal 27 and SCC-9 cells, whereas a substantial downregulation was observed in DDX5-OE Cal 27 and SCC-9 cells (Figure 3E,F). Notably, the knockdown of MMP10 considerably curtailed the viability of DDX5-KD tongue cancer cells, suggesting MMP10 as a pivotal gene that counteracts the DDX5-associated inhibition of tongue cancer cell proliferation (Figure 3G,H).

### 3.7. The Suppression of DDX5 Enhances Chemoresistance in Tongue Cancer Cells

To explore the influence of DDX5 expression levels on the chemotherapy resistance of tongue cancer cells, we extracted the DDX5 expression data pertaining to tongue cancer cell lines from the Cancer Cell Line Encyclopedia (CCLE) database, complemented by IC50 data of several drugs procured from the Genomics of Drug Sensitivity in Cancer (GDSC) database. Utilizing the pRRophetic package, we analyzed the therapeutic responses of tongue cancer cells high in DDX5 and low in DDX5 to the chemotherapy agents gemcitabine, cisplatin, and docetaxel. Remarkably, we found that tongue cancer cells with higher DDX5 expression demonstrated a higher sensitivity to chemotherapy (Figure 4A).

To further delineate the role of DDX5 in modulating resistance to chemotherapy in tongue cancer, we undertook a comparative analysis of cell viability and apoptosis between control and DDX5-KD SCC-9 and Cal 27 cells when exposed to varying doses of cisplatin. We found that the DDX5 knockdown significantly reduced apoptosis rates (Figure 4B,C) and increased cell viability in both SCC-9 and Cal 27 cells under cisplatin treatment (Figure 4D,E). These results indicate that DDX5 negatively regulates tongue cancer cell resistance to chemotherapy.

### 3.8. DDX5 Expression in Tongue Cancer Cells Potentially Modulates Immune Cell Infiltration in the Tumor Microenvironment

We reanalyzed the single-cell RNA sequencing (scRNA-seq) data, GSE172577, deposited in the GEO database [15]. These data encompass analyses of six cases of tongue cancer tissues, facilitated by 10X Genomics technology. Utilizing the Seurat package, we preprocessed the scRNA-seq data, where the PercentageFeatureSet function determined the proportion of mitochondrial genes. Furthermore, a detailed correlation analysis investigated the relationship between sequencing depth and mitochondrial DDX5 sequences, and/or total intracellular sequences. In this process, we established criteria whereby each DDX5 had to be expressed in at least five cells, DDX5 expression per cell ranged between 300 and 5000, mitochondrial content was restricted below 10%, and the UMI per cell was mandated to exceed 1000. Subsequently, the scRNA-seq data were normalized utilizing the LogNormalize method post data filtering.

Employing the run t-distributed stochastic neighbor embedding (TSNE) function, we executed the TSNE dimensionality reduction, facilitating the visualization of individual cells (Figure 5A). Concurrently, the uniform manifold approximation and projection (UMAP) diagram portrayed the distribution patterns of principal cell types, classified according to diverse canonical markers (Figure 5B,C). Utilizing CopyKAT, we were able to discern genomic copy number alterations (CNAs) in the samples obtained from these six patients with tongue cancer, thereby facilitating a nuanced understanding of tumor heterogeneity (Appendix A).

Our analysis extended to evaluating and ranking the expression of DDX5 in individual cancer cells derived from the six tongue cancer tissues under study. A marked variation was observed, with the highest expression of DDX5 documented in patient 4 and the lowest in patient 1 (Figure 5D). Notably, the expression levels between patients 1 and 6 did not exhibit significant discrepancies. However, significant variations were noted when comparing patient 1 with patients 2, 3, 4, or 5, and similarly between patient 6 and patients 2, 3, 4, or 5 (Figure 5D). Extrapolating from these findings, we classified the six samples into two distinct categories: the DDX5-low group (comprising samples from patients 1 and 6) and the DDX5-high group (encompassing samples from patients 2, 3, 4, and 5). Through the application of UMAP plots, we deciphered the cell compositions of the primary cell types present in tongue cancer tissues high in DDX5 and low in DDX5. Intriguingly, our findings illustrated that tumors high in DDX5 exhibited a significantly diminished infiltration of B cells and macrophages compared with tumors low in DDX5. Conversely, an increased infiltration of T cells was evident in tumors high in DDX5 compared with their low-DDX5 counterparts (Figure 5E), indicative of a potential influential role of DDX5 expression in modulating immune cell infiltration in the tumor microenvironment.

### 3.9. DDX5 Expression in Tongue Cancer Cells Favors the Infiltration of T Cell Clusters Exhibiting Anticancer Activities

We meticulously analyzed the distribution of various T-cell clusters in tongue cancer samples high in DDX5 and low in DDX5. Utilizing the indicative classical markers, we classified T cells into distinct clusters: CD4_effector, CD4_follicular helper, CD4_naive, CD4_exhausted, CD8_cytotoxicity, CD8_effector, and CD8_exhausted cells (Figure 6A). We harnessed the capabilities of both UMAP plots, which delineate the absolute numbers of each T cell cluster (Figure 6B), and boxplots, presenting the relative percentage of each T cell cluster (Figure 6C), to discern the comparative distribution between tongue cancer specimens high in DDX5 and those low in DDX5. Our observations revealed a pronounced presence of CD4_effector and CD8_effector cells in tongue cancer tissues high in DDX5, a contrast to the markedly reduced infiltration of cells like CD4_exhausted, CD4_follicular helper, CD4_naive, and CD8_exhausted in the same tissue samples, when compared with tongue cancer tissues low in DDX5 (Figure 6D).

Furthermore, we juxtaposed the average cytotoxic and exhausted signature scores of CD8+ T cells in the respective clusters of tongue cancer samples high in DDX5 and those low in DDX5. Our analysis showed significantly higher cytotoxic signature scores for CD8+ T cells (Figure 6E), alongside significantly diminished exhausted signature scores for the CD8+ T cells (Figure 6F) in tongue cancer tissues high in DDX5 compared with tongue cancer tissues low in DDX5 (Figure 6E,F). Furthermore, utilizing multi-immunofluorescence staining, we analyzed the correlation between DDX5 expression and CD8+ T cell infiltration in clinical tongue cancer tissues. We found that DDX5 expression had a significant correlation with CD8+ T cell infiltration in the TME of tongue cancer tissues (Figure 6G). These results suggest that the high expression of DDX5 in tongue cancer cells favors the infiltration or presentation of these T cell clusters with anticancer function.

### 3.10. DDX5 Expression in Tongue Cancer Cells Is Negatively Correlated with the M2 Macrophage Infiltration in the TME

We further analyzed the macrophage presence in tongue cancer samples high in DDX5 and those low in DDX5. Utilizing the specific markers that indicate diverse functional macrophage subsets, we categorized macrophages into four distinct groups: Macro_apoec3a, Macro_spp1, Macro_thbs1, and Macro_ccnl1. We leveraged both the UMAP plot (Figure 7A) and boxplot (Figure 7B) to examine and compare the infiltration patterns of these macrophage subgroups in tongue cancer tissues high in DDX5 and low in DDX5. Our findings revealed a significant elevation in the presence of Macro_apoec3a macrophages, contrasted by a considerable decrease in the infiltration levels of Macro_spp1, Macro_thbs1, and Macro_ccnl1 macrophages in tissues high in DDX5, when compared with tongue cancer specimens low in DDX5 (Figure 7A,B).

Furthermore, we applied Monocle 2 to visualize the trajectories of DDX5 expression patterns. This allowed us to illustrate the distribution of various macrophage subtypes, distinctly labeled by colors, as they transition through four phases along a pseudo-time continuum (Figure 7C,D).

In addition, we employed multi-immunofluorescence staining to examine the relationship between DDX5 expression and the infiltration of Macro_spp1 (M2 type) in clinical tongue cancer samples. We found a negative association between DDX5 expression and Macro_spp1 (M2 type) infiltration in the TME of tongue cancer tissues (Figure 7E). Collectively, these results indicate that elevated DDX5 expression in tongue cancer potentially inhibits the infiltration of macrophages exhibiting pro-cancer activity.

## 4. Discussion

DDX5 is a multifunctional protein implicated in numerous cellular processes including transcription, translation, and both precursor mRNA and microRNA processing. Many studies have shown that DDX5 is aberrantly expressed and plays an oncogenic role in nearly all types of cancers, including colon cancer [21], breast cancer [22], and prostate cancer [23]. However, our study delineates a contrasting role for DDX5, demonstrating its function as a tumor suppressor in tongue cancer. The high expression of DDX5 in tongue cancer correlates with a better prognosis in clinical patients. Previously, Beier et al. suggested a promotional role for DDX5 in the development of head and neck SCC (HNSCC) cells [24]. In contrast, our study suggests that DDX5 plays a tumor-suppressive role in tongue cancer. Head and neck cancer encompasses a complex array of cancers localized in areas such as the larynx, hypopharynx, oropharynx, and the oral cavity [1], and tongue cancer is one of the most common oral carcinomas. The complexity of head and neck cancer makes it very challenging to compare the data from different reports as they indicate different compositions of tumors from different regions of the head and neck [1].

DDX5 has been perceived as a transcriptional coactivator for several oncogenic transcription factors, including nuclear factor-κβ (NF-κβ) and estrogen receptor α (ER-α) [25]. Remarkably, our research reveals that DDX5 counteracts the expression of genes pivotal to tongue cancer progression, such as MMP10. The inhibition of MMP10 notably diminishes the viability of tongue cancer cells with DDX5 knockdown, suggesting that MMP10 is one of the key genes confronting the DDX5-mediated inhibition of tongue cancer cell proliferation. DDX5 may directly or indirectly influence the transcription of the MMP10 gene. This could involve DDX5 binding to the promoter region of MMP10 or interacting with other transcription factors that regulate MMP10 expression. As an RNA helicase, DDX5 could also play a role in the post-transcriptional modification of MMP10 mRNA. This might include influencing mRNA stability, splicing, or translation efficiency. Experiments could be designed to determine whether DDX5 interacts with MMP10 mRNA or whether it affects its stability and translation. Further experiments will be conducted to validate the interaction mechanism between the DDX5 and MMP10.

Tongue cancer is one the most fatal subtypes of head and neck cancer, and is characterized by rapid growth and development and a high rate of lymph node metastasis [26]. Chemotherapy alone or after surgery is still one of the major therapeutic strategies for tongue cancer, but the development of therapy resistance or de nova insensitivity to chemotherapy is very common. However, the mechanisms underlying drug resistance are largely unknown. The literature reports mutations or deregulations in the EGF receptor, glutathione S transferase (GST), tongue cancer chemotherapy-resistance-associated protein 1 (TCRP1), and zinc finger E-box binding homeobox 1 (ZEB1) as potential contributors to chemotherapy resistance [27]. In this study, our data reveal a novel aspect of DDX5, showing that tongue cancer cells exhibiting higher DDX5 expression levels have increased sensitivity to chemotherapy agents such as gemcitabine and cisplatin. DDX5 knockdown diminishes apoptosis rates and augments cell viability post cisplatin treatment.

In this study, we reanalyzed the scRNA-seq data of six tongue cancers [16]. Our analysis revealed that tumors high in DDX5 exhibited a markedly lower infiltration of B cells and macrophages but a significantly higher T cell infiltration when compared to tumors low in DDX5, indicating that DDX5 potentially modulates the immune/inflammatory cell dynamics in the tumor microenvironment.

CD8+ T cells are at the forefront of the adaptive immune response, bearing crucial responsibility for identifying and eliminating malignant or compromised cells [28]. In the tumor microenvironment, the presence and activity level of these cells often signify the effectiveness of the immune response against tumors. In fact, their infiltration level within tumors often aligns with favorable prognostic outcomes, firmly establishing their role as a positive indicator in numerous cancers [29]. Our study sheds light on a potential correlation between DDX5 expression levels in tumor cells and the extent of CD8+ T cell infiltration. These results suggest that elevated DDX5 expression might foster a TME that promotes CD8+ T cell infiltration, potentially through the orchestrated secretion of specific cytokines or chemokines. On the other hand, decreased DDX5 expression could potentially undermine the TME’s capacity to draw and sustain CD8+ T cells. In addition to orchestrating T cell recruitment, DDX5 might also affect the functional disposition of infiltrating CD8+ T cells. For example, tumors showing lower DDX5 expression might foster a more “exhausted” phenotype in CD8+ T cells, marked by a decrease in cytotoxic fervor and proliferative capabilities [30].

Macrophages are subjected to significant transformations in the dynamic landscape of cancer, highlighting a complex nexus between various macrophage phenotypes and their influence on tumor progression [31]. Initially, M1 macrophages, armed with an array of pro-inflammatory cytokines, might offer a line of defense, potentially thwarting tumor expansion in the nascent stages of cancer development. However, the volatile and frequently adverse TME in cancer can trigger a phenotypic metamorphosis, converting M1 macrophages into the M2 state and subsequently fueling tumor progression [32]. In this study, we demonstrate that high DDX5 levels limit the entry of SPP1 macrophages, manifesting as a deterrent to TAM recruitment. Thus, the fluctuating expression levels of DDX5 are perceived to orchestrate a complex interaction with TAMs, potentially steering the overall trajectory of the disease and influencing prospective therapeutic avenues.

These findings, underscored by single-cell sequencing, are reflected in our immunofluorescence analyses of clinical tongue cancer tissues, revealing an increased proportion of CD8+ T cells and decreased infiltration of M2 macrophages in samples exhibiting high DDX5 expression levels. This intricate synergy between DDX5 expression in tumor cells and CD8+ T cell and/or M2 infiltration creates a promising pathway for innovative therapeutic ventures. It is imperative to further probe the underpinnings of this interaction to potentially exploit it in the design of optimized cancer therapies.

## 5. Conclusions

Our study shows that DDX5 acts as a tumor suppressor and could potentially serve as a valuable prognostic biomarker and therapeutic target in tongue cancer. Enhanced DDX5 expression impedes proliferation, progression, and resistance to cisplatin in tongue cancer and is associated with a more favorable prognosis for clinical patients. Additionally, DDX5 expression in tongue cancer is correlated with the infiltration of CD8+T cells in the TME.

## Figures and Tables

**Figure 1 cancers-15-05882-f001:**
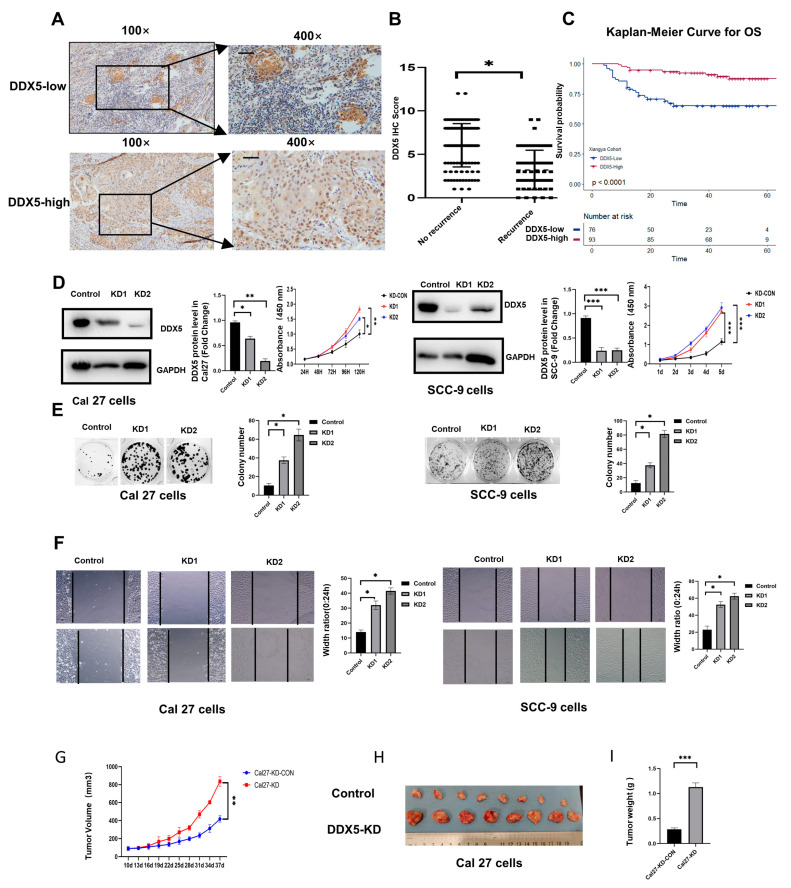
DDX5 downregulation correlates with poor prognosis and contributes to the progression of tongue cancer. (**A**) Representative images of immunohistochemical (IHC) staining for DDX5 protein expression in paraffin-embedded human tongue cancer tissues. Scale bars, 50 mm. (**B**) Correlation of DDX5 protein expression in tongue cancer tissues with tongue cancer relapse. “Non-recurrence” means that tongue cancer had not relapsed after surgery, chemotherapy, and radiation therapy in 3 years. Statistical analysis was performed by using a Mann–Whitney U test in the two groups. (**C**) Overall survival of tongue cancer patients with different DDX5 protein expressions. Log-rank test was utilized to statistically evaluate the disparities in survival curves. (**D**) Knockdown efficiency and proliferation rate of DDX5-KD and control counterpart tongue cancer cells. The uncropped bolts are shown in Appendix A. (**E**) Representative images (left panel) and quantification data (right panel) of cell colony formation assay in DDX5-KD and control counterpart tongue cancer cells. (**F**) Scratch wound healing assay in DDX5-KD and control Cal27 or SCC-9 tongue cancer cells was evaluated. Statistical analysis was performed using one-way ANOVA test in (**D**–**F**). (**G**–**I**) DDX5 knockdown inhibits the growth of Cal27 cells in NOD-SCID mice (**G**). Growth curve (**G**), tumor images (**H**), and weight (**I**). Statistical analysis in (**G**–**I**) was performed using a two-sided unpaired *t*-test. The results are displayed as mean ± SEM. Significance levels are noted as * *p* < 0.05, ** *p* < 0.01, *** *p* < 0.001.

**Figure 2 cancers-15-05882-f002:**
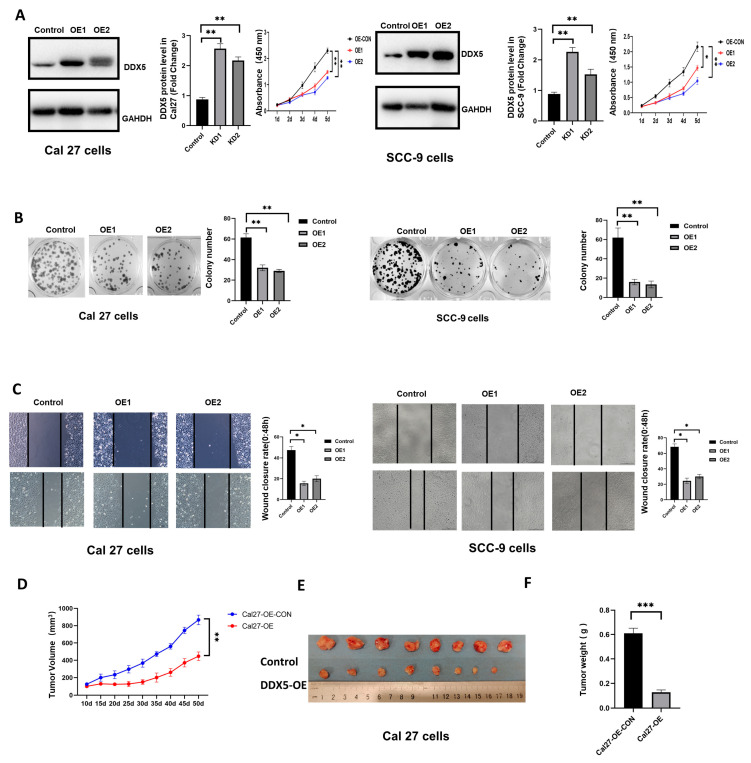
DDX5 overexpression inhibits tongue cancer growth and development. (**A**) The expression of indicated proteins and the proliferation rate of control and stable DDX5 overexpression (DDX5-OE) in Cal27 or SCC-9 tongue cancer cells were analyzed by Western blot and CCK-8 assay. The uncropped bolts are shown in Appendix A. (**B**) Representative images (left panel) and quantification data (right panel) of cell colony formation assay for DDX5-OE and control Cal27 or SCC-9 tongue cancer cells. (**C**) Cell scratch wound healing assay for DDX5-OE and control Cal27 or SCC-9 tongue cancer cells. Statistical analysis was performed using one-way ANOVA test in (**A**–**C**). (**D**–**F**) NOD-SCID mice were inoculated subcutaneously with 5 × 10^6^ DDX5-OE or control Cal27 tongue cancer cells. Tumor development was monitored, and the growth curve was analyzed (**D**). Fifty days after cancer cell inoculation, mice were euthanized and tumors were dissected (**E**) and weighted (**F**). Statistical analysis for (**D**,**F**) was conducted using a two-sided unpaired *t*-test. Data are presented as mean ± SEM. Significance levels are indicated as * *p* < 0.05, ** *p* < 0.01, *** *p* < 0.001.

**Figure 3 cancers-15-05882-f003:**
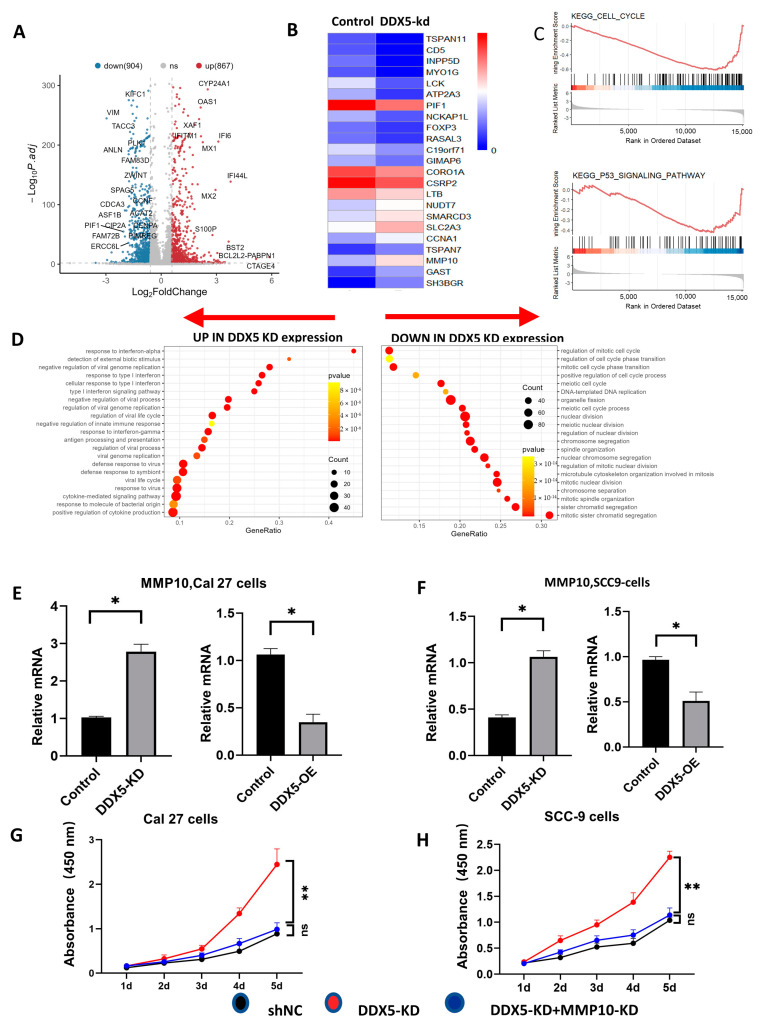
DDX5 decreases the expression of genes related to tumor progression. (**A**) Volcano plot of the differentially expressed genes (DEGs) between DDX5−KD and control counterpart tongue cancer cells. Significant DEGs were identified with an absolute log-fold change > 1.2 and adjusted *p* < 0.05, using a two-sided Wilcoxon rank-sum test and Bonferroni correction. (**B**) Heatmap for DEGs between DDX5−KD and control counterpart cells. (**C**) Gene set enrichment analysis (GSEA) indicates significant enrichment in DDX5−associated pathways. NES (normalized enrichment score) and FDR (false discovery rate). (**D**) Gene ontology (GO) analysis categorizes both upregulated (left) and downregulated (right) genes. The top 21 biological processes involving DEGs are displayed in a dot plot, where dot size denotes the amount of genes involved. Color variation represents the adjusted *p* value for each GO term, and the gene ratio is calculated as the count relative to the total DEGs. (**E**) The expression levels of MMP10 in Cal27 tongue cancer cells, both in control and DDX5−KD, as well as in control and DDX5−OE, were analyzed by using real-time PCR. (**F**) The MMP10 expression in SCC−9 tongue cancer cells was compared between control, DDX5−KD, and DDX5-OE groups by using real-time PCR. A statistical analysis of these comparisons, shown in Cal27 (**E**) and SSC−9 (**F**), was performed with a two-sided unpaired *t*-test. (**G**,**H**) CCK−8 assays were conducted to evaluate proliferation rates in Cal27 (**G**) and SSC−9 (**H**) tongue cancer cell lines following infection with various or combined lentiviruses expressing specific shRNA. A statistical analysis of the results in (**G**,**H**) was performed using a one-way ANOVA test. Data are presented as mean ± SEM. Significance levels are indicated as * *p* < 0.05, ** *p* < 0.01.

**Figure 4 cancers-15-05882-f004:**
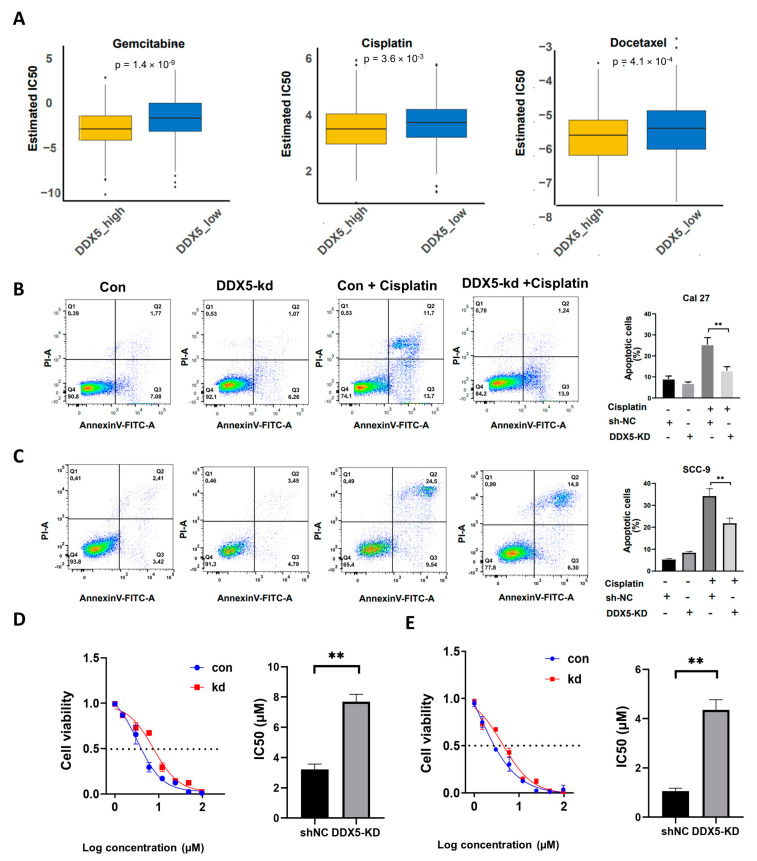
DDX5 downregulation enhances while upregulation decreases cisplatin resistance in tongue cancer cells. (**A**) Boxplots for the putative chemotherapeutic responses of tongue cancer cells to gemcitabine, cisplatin, and docetaxel (data from the CCLE database). Statistical analysis was conducted by using a two-sided unpaired t−test in (**A**). (**B**,**C**) Annexin V/PI and flow cytometry analysis for the connection between cell apoptosis and its DDX5 expression levels in Cal27 (B) or SCC−9 cells (**C**). Statistical analysis was conducted by using one-way ANOVA in (**B**,**C**). (**D**,**E**) CCK-8 assay for the viability of Cal27 (**D**) or SSC−9 (**E**) tongue cancer cells infected with lentiviruses expressing control or DDX5 shRNA, followed 24 h later by treatment with different doses of cisplatin for 48 h. Statistical analysis was conducted by using a two-sided unpaired t test in (**D**,**E**). Results are presented as mean ± SEM. Significance levels are indicated as ** *p* < 0.01.

**Figure 5 cancers-15-05882-f005:**
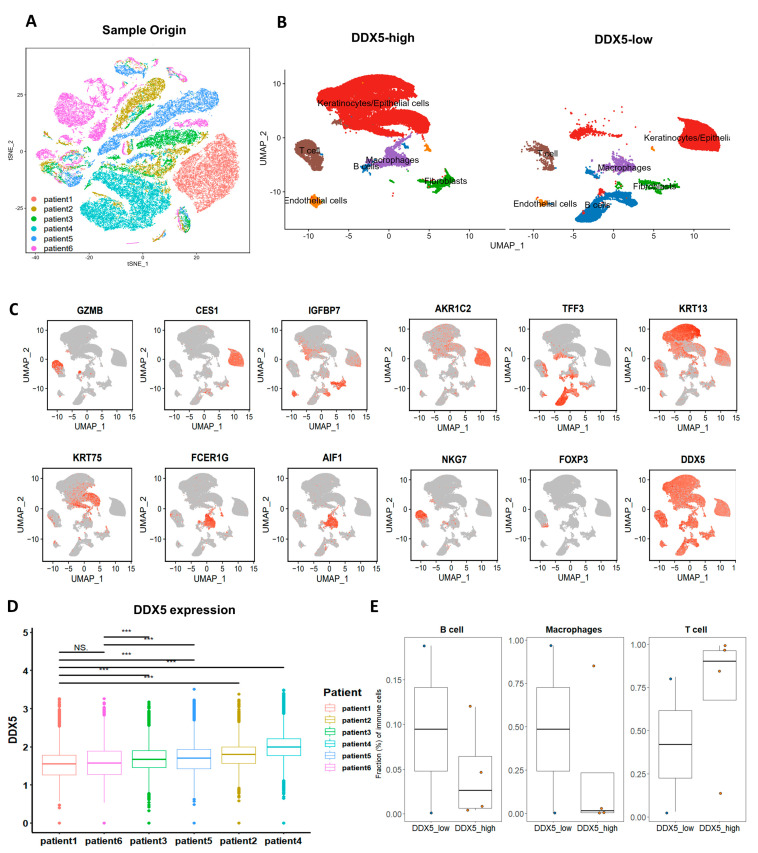
Single−cell analysis in clinical tongue cancer. (**A**) A t−SNE map displaying the cellular distribution of the six tongue cancer samples, with each sample’s cell population indicated by a distinct color. (**B**) The UMAP diagram illustrates the distribution patterns of major cell types, categorized using canonical markers. (**C**) Expression profiles of marker genes for cell types identified in (**B**). (**D**) DDX5 expression levels were compared and ranked in cancer cells from the six different tongue cancer tissues. (**E**) Boxplot depicting the cellular proportions of T cells, macrophages, and B cells in tongue cancer tissues with low DDX5 expression (*n* = 2) versus high DDX5 expression (*n* = 4). In the boxplot, the center line shows the median, while the lower and upper hinges represent the 25th and 75th percentiles, respectively. Whiskers extend to 1.5× the interquartile range. Statistical analysis for (**D**) used the Wilcoxon test and that for (**E**) used the Mann−Whitney U test. Significance levels are indicated as *** *p* < 0.001.

**Figure 6 cancers-15-05882-f006:**
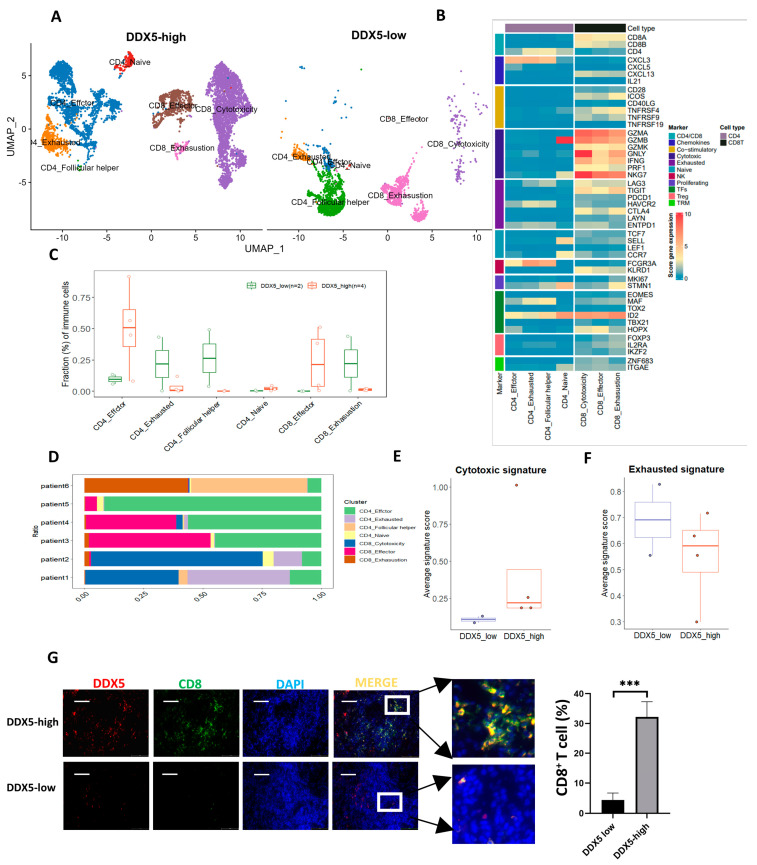
The infiltration of CD8+ T-cell clusters in tongue cancer is associated with the expression levels of DDX5 in the cancer. (**A**) UMAP plot illustrating T−cell clusters, differentiated by various colors, in tongue cancer tissues with low (*n* = 2) and high (*n* = 4) DDX5 expression. (**B**) Heatmap displaying the normalized expression levels of T-cell markers across various clusters. (**C**) Boxplot depicting the proportions of each T-cell cluster in tongue cancer tissues with low (*n* = 2) and high (*n* = 4) DDX5 expression. (**D**) Boxplot illustrating the cellular fractions of each T cluster in individual tongue cancer tissues. Panels (**E**,**F**) show boxplots of average cytotoxic and exhausted signature scores for CD8+ T cells in patients with low DDX5 (*n* = 2) and high DDX5 (*n* = 4) expression. In the boxplots, the center line shows the median, while the lower and upper hinges indicate the 25th and 75th percentiles, respectively. Whiskers extend to 1.5 times the interquartile range. Statistical analysis for panels (**C**,**E**,**F**) was performed using the Mann−Whitney U test. (**G**) The representative images depict multiplex immunofluorescence staining for DDX5 and CD8 in tongue cancer tissues low in DDX5 and high in DDX5. Scale bars, 50 mm. Statistical analysis in (**G**) was conducted using a two-sided unpaired *t*-test. Results are presented as mean ± SEM. Significance levels are indicated as *** *p* < 0.001.

**Figure 7 cancers-15-05882-f007:**
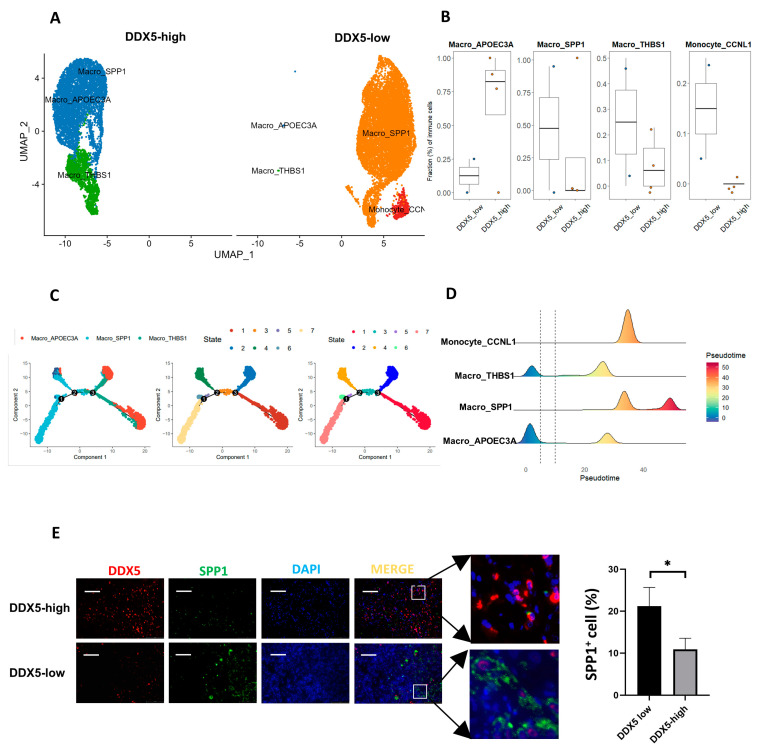
The infiltration of macrophages is associated with DDX5 expression in tongue cancer. (**A**) UMAP plot for macrophages, marked by different colors, in tongue cancer tissues low in DDX5 and high in DDX5. (**B**) Boxplot showing cellular fractions of each macrophage cluster in tongue cancer tissues low in DDX5 (*n* = 2) and high in DDX5 (*n* = 4). Statistical analysis was conducted by using a Mann−Whitney U test in (**B**). (**C**) The trajectories of DDX5 expression were visualized using Monocle 2. (**D**) The distribution of macrophage subtypes during the transition was divided into 4 phases, along with the pseudo−time. Subtypes are labeled by color. (**E**) Representative images of immunofluorescence double-staining for both DDX5 and SPP1 in tongue cancer tissues. Scale bars, 50 mm. Statistical analysis was conducted by using a two−sided unpaired t test in (**E**). Results are presented as mean ± SEM. Significance levels are indicated as * *p* < 0.05.

**Table 1 cancers-15-05882-t001:** Association of DDX5 expression with the clinicopathological characteristics of tongue cancer patients.

Characteristics	DDX5 Expression			
	Overall	Low (*n* = 76)	High (*n* = 93)	*p* Value
Age (*n*, %)				
<50	71(42.0)	31(40.8)	40(43.0)	
≥50	98(58)	45(59.2)	53(57)	0.893
Sex (*n*, %)				
Male	151(89.3)	63(82.9)	88(94.6)	
Female	18(10.7)	13(17.1)	5(5.4)	0.027
T Stage (*n*, %)				
I	62(36.7)	26(34.2)	36(38.7)	
II	72(42.6)	32(42.1)	40(43)	
III	24(14.2)	11(14.5)	13(14)	
IV	11(6.5)	7(9.2)	4(4.3)	0.616
N Stage (*n*, %)				
I	106(62.7)	40(52.6)	66(71.0)	
II	36(21.3)	22(28.9)	14(15.1)	
III	27(16.0)	14(18.4)	13(14.0)	0.038
Metastasis (*n*, %)				
Absence	153(90.5)	63(82.9)	90(96.8)	
Presence	16(9.5)	13(17.1)	3(3.2)	0.005
AJCC Stage (*n*, %)				
I	50(29.6)	16(21.1)	34(36.6)	
II	46(27.2)	19(25)	27(29)	
III	41(24.3)	24(31.6)	17(18.3)	
IV	32(18.9)	17(22.4)	15(16.1)	0.056
Pathology Grade (*n*, %)				
Grade 1	110(65.1)	49(64.5)	61(65.6)	
Grade 2	50(29.6)	21(27.6)	29(31.2)	
Grade 3	9(5.3)	6(7.9)	3(3.2)	0.387

Abbreviations: AJCC: The American Joint Committee on Cancer.

## Data Availability

The datasets used and/or analyzed during the current study are available from the Gene Expression Omnibus (GEO) database upon publication of this study.

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
