# Peer review of "DDX5 Functions as a Tumor Suppressor in Tongue Cancer"

_cancers, 2023, doi:10.3390/cancers15245882_

Round 1

Reviewer 1 Report

Comments and Suggestions for Authors

Qingqing Liu and her collaborators have detailed the role of the DDX5 gene as a tumor suppressor in oral squamous cell carcinoma tumors, effectively inhibiting the proliferation of cancerous cells. This finding holds significant novelty and clinical relevance, especially considering previous studies that suggested an oncogenic role for DDX5. Additionally, the authors have highlighted DDX5's involvement in chemoresistance. The underlying mechanism appears to be linked to the regulation of immune infiltration within the tumor microenvironment (TME).

The manuscript presents a well-addressed study; however, there are some issues that need to be resolved before considering the paper for publication.

-       In the Abstract, authors should consider to change the overstatement: “…DDX5 in tongue cancer is closely related to its ability of regulating immune cells infiltration in tumor microenvironment (TME). DDX5 in tongue cancer decreases the infiltration of M2 macrophages and favors the infiltration of T cell clusters exerting anti-cancer effects in TME.”   

Given that no functional studies with shDDX5 and/or DDX5 OE have been conducted to demonstrate the regulation of immune components by DDX5, and this study has been performed using patient samples with varying levels of DDX5, it would be prudent to rephrase the sentence as follows: "Expression of DDX5 is associated/correlates with reduced infiltration of M2 macrophages and increased infiltration of T cell clusters, which may contribute to anti-cancer effects in the tumor microenvironment."

-       In Material and Methods 2.8, kindly provide precise details regarding the number of tongue cancer cells that were subcutaneously inoculated.

Additionally, it would be valuable for the authors to explain their choice of subcutaneous injection over the more relevant method of injecting cells directly into the oral cavity or tongue, where the tumor could replicate the exact biology of an oral cancer tumor in a patient. This aspect should be discussed in the manuscript.

-       In section 2.10 of the Materials and Methods, please provide a description of the "primary antibodies" along with the appropriate references.

-       Please describe the method to isolate RNA for RNA-sequencing.

-       Figure1A: please label the markers and put the scale bars in the images; Figure 1C, 1F, 2C: consider increase the resolution; Figure 1G, 2D, 3G, 3H: please, label the axis

-       Consider make clearer the title of figure legend: “Figure 3. DDX5 downregulates the expression of DDX5s associated with tongue cancer progression.”

-       Figure 4A: the figure legend indicates treatment with cisplatin, doxorubicin, doxetacel and gemcitabine but there is not plot for doxorubicin treatment.

-       Figure 5A, 5B, Figure 6A, Figure 7A, 7C, 7D: Consider increasing the resolution, as it is difficult to clearly discern the data from the different clusters.

-       Figure 6G, Figure 7E: How the authors established the low/high DDX5 expression in clinical tongue samples?

-       In the discussion, could the authors explore a mechanistic hypothesis regarding the role of the MMP10-DDX5 axis in tongue cancer progression?

Author Response

We sincerely thank the reviewer for their valuable feedback that we have used to improve the quality of our manuscript. The reviewer comments are laid out below in italicized font and specific concerns have been numbered.

Comment 1: In the Abstract, authors should consider to change the overstatement: “…DDX5 in tongue cancer is closely related to its ability of regulating immune cells infiltration in tumor microenvironment (TME). DDX5 in tongue cancer decreases the infiltration of M2 macrophages and favors the infiltration of T cell clusters exerting anti-cancer effects in TME.”  

Given that no functional studies with shDDX5 and/or DDX5 OE have been conducted to demonstrate the regulation of immune components by DDX5, and this study has been performed using patient samples with varying levels of DDX5, it would be prudent to rephrase the sentence as follows: "Expression of DDX5 is associated/correlates with reduced infiltration of M2 macrophages and increased infiltration of T cell clusters, which may contribute to anti-cancer effects in the tumor microenvironment."

Response 1: We gratefully appreciate for your valuable suggestion. Your advice is highly appreciated, and we have accordingly revised the original manuscript.

Comment 2: In Material and Methods 2.8, kindly provide precise details regarding the number of tongue cancer cells that were subcutaneously inoculated.

Additionally, it would be valuable for the authors to explain their choice of subcutaneous injection over the more relevant method of injecting cells directly into the oral cavity or tongue, where the tumor could replicate the exact biology of an oral cancer tumor in a patient. This aspect should be discussed in the manuscript.

Response 2: We gratefully appreciate for your valuable suggestion. In my experiments, I employed a cell count of 5×106, this previously determined through the trials of other team members in our research group.

The orthotopic tumor induction replicates the distinct biological and microenvironmental traits of oral cancer. Consideration of the local impact of oral cancer, including the physiological aspect of the tongue or oral cavity, is feasible.

When I began the preliminary experiments, my mentor recommended that I conduct orthotopic tumor induction in mice. Regrettably, it did not succeed. The likely reasons could be the impact of the mice's oral feeding on tumor fixation and issues with my technique. Hence, we adopted the approach of subcutaneous tumor induction.

Comment 3: In section 2.10 of the Materials and Methods, please provide a description of the "primary antibodies" along with the appropriate references.

Response 3: Thank you for your nice comments. We have added the antibody sources along with the appropriate references to the original manuscript.

DDX5 primary antibodies (Abcam,ab126730) [10]

Xu, J.; Cai, Y.; Ma, Z.; Jiang, B.; Liu, W.; Cheng, J.; Guo, N.; Wang, Z.; Sealy, J.E.; Song, C.; et al. The RNA helicase DDX5 promotes viral infection via regulating N6-methyladenosine levels on the DHX58 and NFκB transcripts to dampen antiviral innate immunity. PLoS Pathog 2021, 17, e1009530, doi:10.1371/journal.ppat.1009530.

Comment 4: Please describe the method to isolate RNA for RNA-sequencing.

Response 4: Thank you for your nice comments. We have added to the original manuscript. “Total RNA was extracted from DDX5-knockdown Cal27 and control cells using the RNeasy Plus Mini Kit (QIAGEN) following the manufacturer's protocol. Following quality control, high-quality total RNA samples with an Agilent Bioanalyzer RIN > 7.0 were employed in the construction of the RNA sequencing library”.

Comment 5: Figure1A: please label the markers and put the scale bars in the images; Figure 1C, 1F, 2C: consider increase the resolution; Figure 1G, 2D, 3G, 3H: please, label the axis

Response 5: Thank you for pointing this out. These have been corrected on the paper.

Comment 6: Consider make clearer the title of figure legend: “Figure 3. DDX5 downregulates the expression of DDX5s associated with tongue cancer progression.”

Response 6: Thank you for pointing this out. The revised text reads as follows that “Figure 3. DDX5 downregulates the expression of genes associated with tongue cancer progression”.

Comment 7: Figure 4A: the figure legend indicates treatment with cisplatin, doxorubicin, doxetacel and gemcitabine but there is not plot for doxorubicin treatment.

Response 7: We were really sorry for our careless mistakes. Thank you for your reminder. The revised text reads as follows that “The box plots for the putative chemotherapeutic responses of tongue cancer cells to gemcitabine, cisplatin and docetaxel”.

Comment 8: Figure 5A, 5B, Figure 6A, Figure 7A, 7C, 7D: Consider increasing the resolution, as it is difficult to clearly discern the data from the different clusters.

Response 8: Thank you for pointing this out. We have made the modifications, enhancing the resolution of the images. Thank you for your suggestion.

Comment 9: Figure 6G, Figure 7E: How the authors established the low/high DDX5 expression in clinical tongue samples?

Response 9Assessment of the patient as a high or low DDX5 expresser is based on the outcomes of immunohistochemical analysis. The interpretation of immunohistochemistry results is scored based on the area of staining and its intensity.

The specific details are following: the score of DDX5 was assessed semi-quantitatively according to the intensity of staining (score 0, no staining; score 1, weak staining; score 2, moderate staining; score 3, strong staining) and the percentage of positive tumor cells (score 0, none; score 1, 1–29%; score 2, 30–69%; score 3, >70%). Multiplying the staining intensity score by the positive tumor cell score generated the overall immunohistochemistry score. The final score ranged from 0 to 9 and was interpreted as follows: negative (0), weak (1–3), moderate (4–6), and strong (>6). NOX4 expression was classified as high (grade 4–9) and low (grade 0–3).

Comment 10: In the discussion, could the authors explore a mechanistic hypothesis regarding the role of the MMP10-DDX5 axis in tongue cancer progression?

Response 10The regulation of MMP10 gene expression by DDX5 can be complex and involves multiple mechanisms. To explore a mechanistic hypothesis, one would need to consider the following potential pathways and interactions.

DDX5 may directly or indirectly influence the transcription of the MMP10 gene. This could involve DDX5 binding to the promoter region of MMP10 or interacting with other transcription factors that regulate MMP10 expression. In addition, DDX5, as an RNA helicase, could also play a role in the post-transcriptional modification of MMP10 mRNA. This might include influencing mRNA stability, splicing, or translation efficiency. Experiments could be designed to determine if DDX5 interacts with MMP10 mRNA or affects its stability and translation. Further experiments will be conducted to validate the interaction mechanism between the DDX5 and MMP10, Including but not limited to chromatin immunoprecipitation (ChIP) and RNA immunoprecipitation sequencing (RIP)

Reviewer 2 Report

Comments and Suggestions for Authors

Dear Jun-Li Luo and colleagues,

I read your manuscript entitled “DDX5 Functions as a Tumor Suppressor in Tongue cancer” with interest. Please see my comments below:

General comment: Although the science is interesting to a wider audience, the terminology used in the text often limits a wider dissemination. Please see my comments below on this matter.

Figure 1A: it is unclear which tissues are shown, please indicate.

Figure 1B Label too close to y-axis

Figure 1G & 2D; Figure 3 G & H unit label is missing at the y-axis

Generally the figure labels are very difficult to read, may be necessary to enlarge them or to break figures up, especially Figures 3A-D; 5A & B; 6A; 7A, C & D

Figure legend 1: please state what the colours red & blue indicate in the legend

P7, L261: please explain to the reader what the three GSE datasets are and why they are important here

Figure 2A overexpressed DDX5 appears to be post-translationally modified – hence the reduction in electrophoretic mobility and the double band – please comment on this

The term “DDXs” is not very clear, better differently expressed genes in DDX5-kd

P9, L324: please give a few examples of up-regulated cytokines especially those once linked with the infiltration of tumour tissue by immune cells as this is very relevant later in the text.

P10, l351: please tell the reader why MMP10 was selected for further analysis (i.e. why is this protein important in this context). Do you have Western blot data on the changes of MMP10 expression? If so, please include.

Figure 4C: please comment on the difference between the Cal27 and SCC-9 cell lines with regard to the block of cell death upon cisplatin treatment in DDX5-kd backgrounds as the effect is much more reduced n SCC-9 cells.

Legend Figure 4A please say that the data come from the CCLE database

P11, L382 onwards: Section 3.8: please introduce the GSE172577 data set and the rational why this particular data set was selected to the reader. In this section two points need to be clarified: (i) what is the control (i.e. normal CCX5 expression) and it needs to be clearly stated whether the presented data are a cross section of all 6 patients or whether the data set of only one patient is presented. Please explain why it is important to know about the mitochondrial DDX5 dependent gene changes (i.e. why is it important to restrict the mitochondrial content to below 10%?).

Related to this: the section om page 12 L392-400 is very difficult to understand for readers who are not aware of the meaning of the technical terms: scRNA-seq data, utilizing the LogNormalize method post data filtering, employing the Run t-Distributed Stochastic Neighbor Embedding (TSNE) function, executed the TSNE dimensionality reduction, the Uniform Manifold Approximation and Projection (UMAP), utilizing CopyKAT. Please introduce them to the reader.

P12, L 397 what are the canonical markers?

P12, L402-404: please explain how the information on the different immune cells can be extracted from the GSE data set. This information comes here “out of the blue”.

Figure 5E not F

P13, l430-431 where do the data n T cell clusters come from? To which DDX-5 high and low samples refers the text.

Figures 6G and Figure 7E labels are missing indicating DDX5 status

P15, L461-465: please explain were the data n DC8+ T cells come from (again this information appears out of the box)

P15, L471: please introduce Macro_spp1, Macro_thbs1, and Macro_ccnl1 ti the reader. Please explain what these labels mean and why they are important here. Say why the genes are of interest (i.e. spp1).

P15, L478: please introduce Monocle2 to the reader

At the beginning of the section starting on line 482, please introduce Spp1 here to the reader. If you have western blot data on Spp1 protein levels, please include.

P17, L516: please tell the reader more on MMP10 and its importance in malignant transformation

P17, L534: please tell the reader why the decline in macrophages & B cells is important

P17, L539: Tumour Micro Environment (TME)

P17, L546: please include examples of cytokines expression of which is altered upon DDX-5 expression that are relevant to the migration of immune cells.

P18, L560 onwards: it might be better to say that high DDX-5 levels limit entry of SPP1 macrophages as this would fit with the data shown in Figure 5E

Comments on the Quality of English Language

Enlish needs only minor editing (but a better narrative :))

Author Response

We feel great thanks for your professional review work on our article. As you are concerned, there are several problems that need to be addressed. According to your nice suggestions, we have made extensive corrections to our previous draft, the detailed corrections are listed below.

Comment 1: Figure 1A: it is unclear which tissues are shown, please indicate.

Response 1: Thank you for pointing this out. We have already added in the figure caption section.

Comment 2: Figure 1B Label too close to y-axis

Response 2: Thank you for pointing this out.We were really sorry for our careless mistakes. Thanks for your correction.

Comment 3: Figure 1G & 2D; Figure 3 G & H unit label is missing at the y-axis

Response: Thank you for pointing this out.We have carefully checked the manuscript and corrected the errors accordingly.

Comment 4: Generally the figure labels are very difficult to read, may be necessary to enlarge them or to break figures up, especially Figures 3A-D; 5A & B; 6A; 7A, C & D

Response 5: Thank you for pointing this out. We have carefully checked the manuscript and corrected the errors accordingly.

Comment 5: Figure legend 1: please state what the colours red & blue indicate in the legend

Response 5: Thank you for pointing this out. Red represents patients with high DDX5 expression and Blue represents patients with low DDX5 expression

Comment 6: P7, L261: please explain to the reader what the three GSE datasets are and why they are important here

Response: 6: Thank you for your nice comments. The dataset comprises normal tongue tissue, carcinoma in situ of the tongue, and tongue cancer, charting the disease's evolution, within this trend, it is observable that DDX5 expression diminishes from carcinoma in situ to the progression into cancer. In previous studies, most tumor research has only compared the difference of DDX5 between tumors and normal tissues, without considering the changes of DDX5 during tumor progression. Based on existing literature and some experimental results, we speculate that inflammatory factors may play a significant role in the transformation of normal tongue tissue to dysplasia, with ddx5 expression possibly increasing initially, but decreasing as it progresses to tumor.

Comment 7: Figure 2A overexpressed DDX5 appears to be post-translationally modified – hence the reduction in electrophoretic mobility and the double band – please comment on this

Response 7: Thank you for your nice comments. HA tag was added to the DDX5 overexpression plasmid, resulting in protein bands that are higher than the original bands after overexpression. Regarding the protein modifications you mentioned, it is indeed found in other literature that DDX5 is modified by sumo, affecting the protein's stability. However, I have not conducted such tests in my experiments.

Comment 8: The term “DDXs” is not very clear, better differently expressed genes in DDX5-kd

Response7: Thank you for pointing this out. These have been corrected on the paper

Comment 8: P9, L324: please give a few examples of up-regulated cytokines especially those once linked with the infiltration of tumour tissue by immune cells as this is very relevant later in the text.

Response8: Thank you for your nice comments. In our RNA sequence data, increased expression of CCL17 is observed in tongue cancer cells with DDX5 knockdown. There is a close association between CCL17 and M2 macrophages. Elevated levels of CCL22 and CCL5 are found in DDX5-knockdown tongue cancer cells. These two cytokines are associated with T cell function.

Comment 9: P10, l351: please tell the reader why MMP10 was selected for further analysis (i.e. why is this protein important in this context). Do you have Western blot data on the changes of MMP10 expression? If so, please include.

Response 9: Based on transcriptomic data, we selected genes with significant differential expression, designed primers, and performed qPCR validation, finding that MMP10 was the gene with the most significant change. We have not yet performed validation using western blot bands. This is primarily because the sequencing results revealed a significant downregulation of MMP10, so using primers to validate RNA levels is the most direct and relevant approach. DDX5 is associated with RNA metabolism and affects the stability of mRNA in downstream genes.

Thank you for your suggestions and we will enhance our experiments accordingly. Other reviewers have also raised similar questions, and we need to further validate the interaction mechanism between DDX5 and MMP10.

Comment 10: Figure 4C: please comment on the difference between the Cal27 and SCC-9 cell lines with regard to the block of cell death upon cisplatin treatment in DDX5-kd backgrounds as the effect is much more reduced n SCC-9 cells.

Response 10: Thank you for your nice comments. Cal27 and SCC-9 cell lines, derived from different types of oral cancers (adenosquamous carcinoma and squamous carcinoma, respectively), inherently possess distinct biological and genetic characteristics. These inherent differences can affect how they respond to treatments like cisplatin.

Comment 11: Legend Figure 4A please say that the data come from the CCLE database

Response 11: Thank you for pointing this out. These have been corrected on the paper.

Comment 12P11, L382 onwards: Section 3.8: please introduce the GSE172577 data set and the rational why this particular data set was selected to the reader. In this section two points need to be clarified: (i) what is the control (i.e. normal CCX5 expression) and it needs to be clearly stated whether the presented data are a cross section of all 6 patients or whether the data set of only one patient is presented. Please explain why it is important to know about the mitochondrial DDX5 dependent gene changes (i.e. why is it important to restrict the mitochondrial content to below 10%?).

Response 12: Thank you for your nice comments. High mitochondrial gene expression in scRNA-seq data often indicates cellular stress or apoptosis. Cells with a higher percentage of mitochondrial transcripts (usually above 10%) are typically considered damaged or dying. By restricting the analysis to cells with lower mitochondrial content, researchers can focus on viable and functionally representative cells, thereby obtaining more accurate and biologically relevant data.

Comment 13Related to this: the section om page 12 L392-400 is very difficult to understand for readers who are not aware of the meaning of the technical terms: scRNA-seq data, utilizing the LogNormalize method post data filtering, employing the Run t-Distributed Stochastic Neighbor Embedding (TSNE) function, executed the TSNE dimensionality reduction, the Uniform Manifold Approximation and Projection (UMAP), utilizing CopyKAT. Please introduce them to the reader.

Response 13: Thank you for your nice comments. We have introduced these key terms in detail in the methods section.

LogNormalize Method: This is a data normalization technique used in scRNA-seq data analysis. It typically involves transforming raw read counts to log-scale, which helps in mitigating the effects of technical variation and amplifying the biological signals. This method is essential for comparing gene expression levels across different cells or samples.

Run t-Distributed Stochastic Neighbor Embedding (TSNE): TSNE is a popular dimensionality reduction technique used to visualize high-dimensional data, like scRNA-seq data, in two or three dimensions. This method helps in identifying patterns, clusters, or groups within the data by preserving the local structure and relative distances between points (cells).

TSNE Dimensionality Reduction: This refers to the application of the TSNE method specifically for reducing the dimensionality of scRNA-seq data. It's particularly useful in visualizing complex cellular heterogeneity and identifying distinct cell populations within a sample.

Uniform Manifold Approximation and Projection (UMAP): UMAP is another dimensionality reduction technique often used in scRNA-seq data analysis. It's similar to TSNE but often faster and better at preserving both the local and global data structure. UMAP provides a more comprehensive view of the data's manifold structure, making it easier to interpret biological variations.

CopyKAT: CopyKAT (Copy number Karyotyping of Aneuploid Tumors) is a computational method used to infer genomic copy number profiles and identify aneuploid cells directly from scRNA-seq data. This tool is particularly valuable in cancer research, where it can help distinguish between normal and cancerous cells based on their genomic copy number alterations.

Comment 14P12, L 397 what are the canonical markers?

Response 14: In single-cell sequencing, especially in single-cell RNA sequencing (scRNA-seq), canonical markers are specific genes that are characteristically expressed in certain cell types or cell states. These markers are crucial for identifying and classifying distinct cell populations within a heterogeneous sample.

T cells: CD3D, CD3E, CD4, CD8

B cells: CD19, MS4A1, CD74;

Macrophages:APOC1, SPP1 and APOE

Fibroblasts: COL1A2, COL3A1 and COL1A1);

Epithelial cells:PECAM1 and A2M

Keratinocytes: S100AB, KRT16 and SPRR3

Comment 15P12, L402-404: please explain how the information on the different immune cells can be extracted from the GSE data set. This information comes here “out of the blue”.

Response 15: Using the Seurat package for dimensionality reduction and clustering in single-cell sequencing allows for the identification of distinct cell populations, Cell populations are annotated according to the expression of marker genes in each group, allowing for the immune cell groups to be extracted separately for further analysis. Given previous findings that DDX5 is crucial in antiviral immunity, and its ability to promote the secretion of inflammatory factors, we considered that in tongue cancers with different levels of DDX5 expression, there might be variations in immune function, leading us to perform this analysis.

Comment 16Figure 5E not F

Response 16: Thank you for pointing this out. These have been corrected on the paper.

Comment 17P13, l430-431 where do the data n T cell clusters come from? To which DDX-5 high and low samples refers the text.

Response 17: T cell clusters were analyzed from single-cell sequencing data of 6 tongue cancer patients. Using the Seurat package for dimensionality reduction and clustering in single-cell sequencing allows for the identification of distinct cell populations, Cell populations are annotated according to the expression of marker genes in each group, allowing for the immune cell groups to be extracted separately for further analysis. The subgroups identified as T cells were extracted for further analysis.

High and low DDX5 expression levels are delineated in section 5 of the figure, based on the expression of DDX5 in tumor cells identified in the single-cell data, they are divided into two groups, groups 1 and 6 are categorized as DDX5_low due to low DDX5 expression, and groups 2, 3, 4, 5 as DDX5_high due to high DDX5 expression.

Comment 18Figures 6G and Figure 7E labels are missing indicating DDX5 status

Response 18: Thank you for pointing this out. These have been corrected on the paper.

Comment 19P15, L461-465: please explain were the data n DC8+ T cells come from (again this information appears out of the box)

Response 19:  CD8+ T cell clusters were analyzed from single-cell sequencing data of 6 tongue cancer patients. The Seurat package was used for dimensionality reduction and clustering in single-cell sequencing, allowing for the identification of distinct cell populations. Cell populations are annotated according to the expression of marker genes in each group, allowing for the immune cell groups to be extracted separately for further analysis. The subgroups identified as T cells were extracted for further analysis. CD8+ T cell clusters were analyzed from the subgroups identified as T cells.

Comment 20P15, L471: please introduce Macro_spp1, Macro_thbs1, and Macro_ccnl1 ti the reader. Please explain what these labels mean and why they are important here. Say why the genes are of interest (i.e. spp1).

Response 20: Spp1 is the most extensively studied gene related to macrophages, and in other literature, spp1+ macrophages are more similar to M2-type macrophages.

Comment 21P15, L478: please introduce Monocle2 to the reader

Response 21: Monocle 2 is an advanced bioinformatics tool designed for analyzing single-cell RNA sequencing (scRNA-seq) data. It is particularly renowned for its ability to order cells based on their progression through biological processes, such as differentiation or disease progression.

Comment 22At the beginning of the section starting on line 482, please introduce Spp1 here to the reader. If you have western blot data on Spp1 protein levels, please include.

Response 22: Western blot analysis for Spp1 expression was not conducted. Spp1 is mostly expressed in immune cells, particularly in M2-type macrophages. In tongue cancer samples with high DDX5 expression, there is a reduction in spp1 expressing cells. Our ongoing cell migration experiments indicate that knocking down DDX5 in tongue cancer cells can increase the induction of M1 cell migration to tumor cells. We will use more indicators to verify whether spp1 originates from macrophages (cd68+).

Thank you for your suggestion.

Comment 23P17, L516: please tell the reader more on MMP10 and its importance in malignant transformation:

Response 23: We sincerely appreciate the valuable comments. MMP10 is involved in the degradation of the extracellular matrix (ECM), which is a critical step in tumor progression. By breaking down ECM components, MMP10 facilitates tumor invasion and metastasis, allowing cancer cells to spread to new sites. MMP10 plays a role in cancer cell migration, a key aspect of cancer metastasis. Its activity is linked to the ability of cancer cells to move from the primary tumor site to other parts of the body.

Comment 24:P17, L534: please tell the reader why the decline in macrophages & B cells is important

Response 24: Thank you for your nice comments. Macrophages can differentiate into tumor-associated macrophages (TAMs) in the tumor microenvironment. TAMs often adopt a pro-tumorigenic phenotype (M2-like macrophages) that supports tumor growth, angiogenesis (formation of new blood vessels), and metastasis. A decline in macrophages could suggest a reduction in the pro-tumorigenic M2-like TAMs, potentially hindering the tumor's ability to grow and spread. Conversely, it could also indicate a loss of potential anti-tumor macrophage activity, which could be detrimental in controlling tumor progression.

B cells can contribute to anti-tumor immunity through the production of antibodies, presentation of tumor antigens to T cells, and secretion of cytokines. They can help in forming tertiary lymphoid structures within tumors, which are associated with improved immune response and prognosis. A decrease in B cells within a tumor could lead to reduced antibody-mediated tumor cell targeting, decreased antigen presentation, and a generally weaker adaptive immune response against the tumor.

Comment 25P17, L539: Tumour Micro Environment (TME)

Response 25:Thank you for pointing this out. These have been corrected on the paper.

Comment 26P17, L546: please include examples of cytokines expression of which is altered upon DDX-5 expression that are relevant to the migration of immune cells.

Response 26: Thank you for your nice comments. This comment is similar to comment 8. In our RNA sequence data, increased expression of CCL17 is observed in tongue cancer cells with DDX5 knockdown. There is a close association between CCL17 and M2 macrophages. Elevated levels of CCL22 and CCL5 are found in DDX5-knockdown tongue cancer cells. These two cytokines are associated with T cell function.

Comment 27P18, L560 onwards: it might be better to say that high DDX-5 levels limit entry of SPP1 macrophages as this would fit with the data shown in Figure 5E

Response 27We sincerely appreciate the valuable comments. These have been corrected on the paper.

Reviewer 3 Report

Comments and Suggestions for Authors

Major concerns:

1. Page 4. The titles of subsections 2.11 and 2.12 are identical.  Please, change into different titles or combine these two sections into a new one.

2. Multipart figure 1 on page 6. Legend to the figure - provide information (in parentheses) about statistical tests used to analyze the data. For example: (Student’s t-test: *P<0.05) or (Kruskal-Wallis test: ***P<0.001)

3. Line 256. The number of DDX5-KD cells. Please, confirm that the number of cell was 5×106. Perhaps, 5×106 ?

4. Multipart figure 2 on page 8. Legend to the figure. Add explanation for (***) 3 asterisks present on figure F. Add information about statistical tests used to analyze the data.

5. Legend to the figure 3 on page 10. Add information about statistical tests used to analyze the data and add explanation for asterisks present on figure.

6. Legend to the figure 4 on page 11. Add information about statistical tests used to analyze the data and add explanation for asterisks present on figure.

7. Line 383 – please change the citation [Peng et al., 2021] according to the journal’s requirement.

8. Please describe briefly the criteria for selecting 6 tongue cancer tissues out of the 169 formalin-fixed tissue samples (page 4, section 2.9). Why only 6 patients were chosen for the analysis of DDX5 expression? Why these patients were selected? What if other samples were selected. Who selected the tissue samples - expert in pathomorphology? 

9. Page 4. Line 154. The Table 1 is missing. It is crucial for Figure 5. Provide clinical and pathological parameters of the patients, otherwise, the results cannot be reliable. How many patients underwent examination (from June 200 to December 2016)? 

10. Legend to the figure 6 on page 15. Add information about statistical tests used to analyze the data and add explanation for asterisks present on figure G.

11. Legend to the figure 7 on page 16. Add information about statistical tests used to analyze the data and add explanation for asterisks present on the column graph (Figure E).

12. Description of statistical tests used in this study MUST be presented in the last subsection of the Materials and Methods.

13. Please, increase the resolution of the multipart-figures to the highest possible values.  Now, illustrations are of low quality that disturb to understand the results presented by the authors.

Comments on the Quality of English Language

No specific comments

Author Response

We feel great thanks for your professional review work on our article. As you are concerned, there are several problems that need to be addressed. According to your nice suggestions, we have made extensive corrections to our previous draft, the detailed corrections are listed below.

Comment 1: Page 4. The titles of subsections 2.11 and 2.12 are identical.  Please, change into different titles or combine these two sections into a new one.

Response 1: Thank you for pointing this out. These have been corrected on the paper.

Comment 2: Multipart figure 1 on page 6. Legend to the figure - provide information (in parentheses) about statistical tests used to analyze the data. For example: (Student’s t-test: *P<0.05) or (Kruskal-Wallis test: ***P<0.001)

p values were calculated using two-sided unpaired t test.

Response 2: Thank you for pointing this out. These have been corrected on the paper. These have been corrected into “Statistical analysis was calculated using one-way ANOVA and Dunnett’s multiple comparison test in (D, E and F) and performed using two-sided unpaired t test in (G-I). Results are presented as mean ± SEM. *P<0.05, **P<0.01, ***P<0.01”.

Comment 3: Line 256. The number of DDX5-KD cells. Please, confirm that the number of cell was 5×106. Perhaps, 5×106 ?

Response 3: Thank you for pointing this out. These have been corrected on the paper.

Comment 4: Multipart figure 2 on page 8. Legend to the figure. Add explanation for (***) 3 asterisks present on figure F. Add information about statistical tests used to analyze the data.

Response 4: Thank you for pointing this out. These have been corrected on the paper.

Comment 5: Legend to the figure 3 on page 10. Add information about statistical tests used to analyze the data and add explanation for asterisks present on figure.

Response 5: Thank you for pointing this out. These have been corrected on the paper.

Comment 6: Legend to the figure 4 on page 11. Add information about statistical tests used to analyze the data and add explanation for asterisks present on figure.

Response 6: Thank you for pointing this out. These have been corrected on the paper.

Comment 7: Line 383 – please change the citation [Peng et al., 2021] according to the journal’s requirement.

Response 7: Thank you for pointing this out. These have been corrected on the paper.

Comment 8: Please describe briefly the criteria for selecting 6 tongue cancer tissues out of the 169 formalin-fixed tissue samples (page 4, section 2.9). Why only 6 patients were chosen for the analysis of DDX5 expression? Why these patients were selected? What if other samples were selected. Who selected the tissue samples - expert in pathomorphology? 

Response 8: Thank you for your nice comments. Our study employed single-cell sequencing data for tongue cancer from the GEO database, not from 169 patients with paraffin-embedded samples. Based on the expression levels of DDX5 (as shown in Figure 5D), patients were divided into two groups: DDX5-high and DDX5-low expression. Patient 1 and 6 were in one group, and 2, 3, 4, 5 in another, exhibiting statistically significant expression differences. All six tongue cancer patients in the sequencing study were diagnosed by pathologists.

Comment 9: Page 4. Line 154. The Table 1 is missing. It is crucial for Figure 5. Provide clinical and pathological parameters of the patients, otherwise, the results cannot be reliable. How many patients underwent examination (from June 200 to December 2016)? 

Response 9: Thank you for pointing this out. These have been corrected on the paper. The clinical information table of tongue cancer patients has been added into the article.

Comment 10: Legend to the figure 6 on page 15. Add information about statistical tests used to analyze the data and add explanation for asterisks present on figure G.

Response 10: Thank you for pointing this out. These have been corrected on the paper.

Comment 11: Legend to the figure 7 on page 16. Add information about statistical tests used to analyze the data and add explanation for asterisks present on the column graph (Figure E).

Response 11: Thank you for pointing this out. These have been corrected on the paper.

Comment 12: Description of statistical tests used in this study MUST be presented in the last subsection of the Materials and Methods.

Response 12: Thank you for pointing this out. These have been corrected on the paper. The description of statistical tests s has been added into the article.

Comment 13: Please, increase the resolution of the multipart-figures to the highest possible values.  Now, illustrations are of low quality that disturb to understand the results presented by the authors.

Response 13: Thank you for pointing this out. These have been corrected on the paper.

Reviewer 4 Report

Comments and Suggestions for Authors

Thank you for the opportunity reviweing your article. I could read this very interesting.

I found some mistakes. Plsase check and revice.

Table 1 is missed.

Figure 1G, Figure 2 D, Figure3G, H: scale legend of vertical axis “Absorbance (450nm) is missed.

Figure 1 B and C are covered.

Figure 4 Gemcitabine is not administered for oral cancer. Why gemcitabine is added to this study?

L260-269 Figure S1D is missed in present.

Comments on the Quality of English Language

Almost good.

Author Response

Thank you again for your positive comments and valuable suggestions to improve the quality of our manuscript.

Comment 1: Table 1 is missed.

Response1: Thank you for pointing this out. These have been corrected on the paper.

Comment 2: Figure 1G, Figure 2 D, Figure3G, H: scale legend of vertical axis “Absorbance (450nm) is missed.

Response 2: Thank you for pointing this out. These have been corrected on the paper.

Comment 3: Figure 1 B and C are covered.

Response 3: Thank you for pointing this out. These have been corrected on the paper.

Comment 4: Figure 4 Gemcitabine is not administered for oral cancer. Why gemcitabine is added to this study?

Response 4:  Thank you for pointing this out. While gemcitabine is commonly used for pancreatic, breast, ovarian, and non-small cell lung cancers, researchers might be exploring its effectiveness in treating oral cancer. Your question is very pertinent, indicating a significant future research topic, since DDX5 expression is not checked in clinical settings for tongue cancer patients receiving chemotherapy. From a basic experimental research perspective, I am highlighting that high DDX5 expression could potentially alter sensitivity to commonly used chemotherapy drugs in clinical practice. Variations in DDX5 expression influence the sensitivity of tongue cancer cells. This could also provide valuable insights for the treatment of other tumor types.

Response 5: L260-269 Figure S1D is missed in present.

Response 5: Thank you for pointing this out. These have been corrected on the paper.

Round 2

Reviewer 2 Report

Comments and Suggestions for Authors

Dear authors,

thank you very much for your thorough response to my comments - this is much appreciated. The manuscript is now much improved and reads more fluently for colleagues who are not familiar with this field of research..

Comments on the Quality of English Language

Moderate English editing may be needed to enhance text flow

Author Response

Thank you again for your positive comments and valuable suggestions to improve the quality of our manuscript.

Reviewer 3 Report

Comments and Suggestions for Authors

No further comments

Comments on the Quality of English Language

English style and grammar must be edited by a person fluent in English

line 333. Mouse models was used ....? it should be: ...were used - plural form!

Author Response

We sincerely thank the reviewer for their valuable feedback that we have used to improve the quality of our manuscript. The reviewer comments are laid out below in italicized font and specific concerns have been numbered.

Comment 1: line 333. Mouse models was used ....? it should be: ...were used - plural form!

Response 1: Thank you for pointing this out. These have been corrected on the paper.